# Stochastic Gradient Descent, Weighted Sampling, and the Randomized Kaczmarz algorithm

**Deanna Needell**
Department of Mathematical Sciences
Claremont McKenna College
Claremont CA 91711
dneedell@cmc.edu

**Nathan Srebro**
Toyota Technological Institute at Chicago
*and* Dept. of Computer Science, Technion
nati@ttic.edu

**Rachel Ward**
Department of Mathematics
Univ. of Texas, Austin
rward@math.utexas.edu

## Abstract

We improve a recent guarantee of Bach and Moulines on the linear convergence of SGD for smooth and strongly convex objectives, reducing a quadratic dependence on the strong convexity to a linear dependence. Furthermore, we show how reweighting the sampling distribution (i.e. importance sampling) is necessary in order to further improve convergence, and obtain a linear dependence on average smoothness, dominating previous results, and more broadly discus how importance sampling for SGD can improve convergence also in other scenarios. Our results are based on a connection between SGD and the *randomized Kaczmarz algorithm*, which allows us to transfer ideas between the separate bodies of literature studying each of the two methods.

## 1 Introduction

This paper concerns two algorithms which until now have remained somewhat disjoint in the literature: the randomized Kaczmarz algorithm for solving linear systems and the stochastic gradient descent (SGD) method for optimizing a convex objective using unbiased gradient estimates. The connection enables us to make contributions by borrowing from each body of literature to the other. In particular, it helps us highlight the role of weighted sampling for SGD and obtain a tighter guarantee on the linear convergence regime of SGD.

Our starting point is a recent analysis on convergence of the SGD iterates. Considering a stochastic objective $F(\boldsymbol{x}) = \mathbb{E}_i[f_i(\boldsymbol{x})]$, classical analyses of SGD show a polynomial rate on the suboptimality of the objective value $F(\boldsymbol{x}_k) - F(\boldsymbol{x}_\star)$. Bach and Moulines [1] showed that if $F(\boldsymbol{x})$ is $\mu$-strongly convex, $f_i(\boldsymbol{x})$ are $L_i$-smooth (i.e. their gradients are $L_i$-Lipschitz), and $\boldsymbol{x}_\star$ is a minimizer of (almost) all $f_i(\boldsymbol{x})$ (i.e. $\mathbb{P}_i(\nabla f_i(\boldsymbol{x}_\star) = 0) = 1$), then $\mathbb{E}\|\boldsymbol{x}_k - \boldsymbol{x}_\star\|$ goes to zero exponentially, rather then polynomially, in $k$. That is, reaching a desired accuracy of $\mathbb{E}\|\boldsymbol{x}_k - \boldsymbol{x}_\star\|^2 \le \varepsilon$ requires a number of steps that scales only logarithmically in $1/\varepsilon$. Bach and Moulines's bound on the required number of iterations further depends on the average *squared* conditioning number $\mathbb{E}[(L_i/\mu)^2]$.

In a seemingly independent line of research, the Kaczmarz method was proposed as an iterative method for solving overdetermined systems of linear equations [7]. The simplicity of the method makes it popular in applications ranging from computer tomography to digital signal processing [5,

9, 6]. Recently, Strohmer and Vershynin [19] proposed a variant of the Kaczmarz method which selects rows with probability proportional to their squared norm, and showed that using this selection strategy, a desired accuracy of $\varepsilon$ can be reached in the noiseless setting in a number of steps that scales with $\log(1/\varepsilon)$ and only *linearly* in the condition number. As we discuss in Section 5, the randomized Kaczmarz algorithm is in fact a special case of stochastic gradient descent.

Inspired by the above analysis, we prove improved convergence results for generic SGD, as well as for SGD with gradient estimates chosen based on a weighted sampling distribution, highlighting the role of importance sampling in SGD:

We first show that without perturbing the sampling distribution, we can obtain a linear dependence on the *uniform conditioning* $(\sup L_i/\mu)$, but it is not possible to obtain a linear dependence on the *average conditioning* $\mathbb{E}[L_i]/\mu$. This is a quadratic improvement over [1] in regimes where the components have similar Lipschitz constants (Theorem 2.1 in Section 2).

We then show that with weighted sampling we can obtain a linear dependence on the average conditioning $\mathbb{E}[L_i]/\mu$, dominating the quadratic dependence of [1] (Corollary 3.1 in Section 3).

In Section 4, we show how also for smooth but not-strongly-convex objectives, importance sampling can improve a dependence on a uniform bound over smoothness, $(\sup L_i)$, to a dependence on the average smoothness $\mathbb{E}[L_i]$—such an improvement is not possible without importance sampling. For non-smooth objectives, we show that importance sampling can eliminate a dependence on the variance in the Lipschitz constants of the components.

Finally, in Section 5, we turn to the Kaczmarz algorithm, and show we can improve known guarantees in this context as well.

## 2    SGD for Strongly Convex Smooth Optimization

We consider the problem of minimizing a strongly convex function of the form $F(\boldsymbol{x}) = \mathbb{E}_{i\sim\mathcal{D}}f_i(\boldsymbol{x})$ where $f_i : \mathcal{H} \to \mathcal{R}$ are smooth functionals over $\mathcal{H} = \mathcal{R}^d$ endowed with the standard Euclidean norm $\|\cdot\|_2$, or over a Hilbert space $\mathcal{H}$ with the norm $\|\cdot\|_2$. Here $i$ is drawn from some *source distribution* $\mathcal{D}$ over an arbitrary probability space. Throughout this manuscript, unless explicitly specified otherwise, expectations will be with respect to indices drawn from the source distribution $\mathcal{D}$. We denote the unique minimum $\boldsymbol{x}_\star = \arg\min F(\boldsymbol{x})$ and denote by $\sigma^2$ the "residual" quantity at the minimum, $\sigma^2 = \mathbb{E}\|\nabla f_i(\boldsymbol{x}_\star)\|_2^2$.

**Assumptions**  Our bounds will be based on the following assumptions and quantities: First, $F$ has strong convexity parameter $\mu$; that is, $\langle \boldsymbol{x} - \boldsymbol{y}, \nabla\boldsymbol{F}(\boldsymbol{x}) - \nabla\boldsymbol{F}(\boldsymbol{y})\rangle \geq \mu\|\boldsymbol{x} - \boldsymbol{y}\|_2^2$ for all vectors $\boldsymbol{x}$ and $\boldsymbol{y}$. Second, each $f_i$ is continuously differentiable and the gradient function $\nabla f_i$ has Lipschitz constant $L_i$; that is, $\|\nabla f_i(\boldsymbol{x}) - \nabla f_i(\boldsymbol{y})\|_2 \leq L_i\|\boldsymbol{x} - \boldsymbol{y}\|_2$ for all vectors $\boldsymbol{x}$ and $\boldsymbol{y}$. We denote $\sup L$ the supremum of the support of $L_i$, i.e. the smallest $L$ such that $L_i \leq L$ a.s., and similarly denote $\inf L$ the infimum. We denote the average Lipschitz constant as $\overline{L} = \mathbb{E}L_i$.

An unbiased gradient estimate for $F(\boldsymbol{x})$ can be obtained by drawing $i \sim \mathcal{D}$ and using $\nabla f_i(\boldsymbol{x})$ as the estimate. The SGD updates with (fixed) step size $\gamma$ based on these gradient estimates are given by:

$$\boldsymbol{x}_{k+1} \leftarrow \boldsymbol{x}_k - \gamma\nabla f_{i_k}(\boldsymbol{x}_k) \tag{2.1}$$

where $\{i_k\}$ are drawn i.i.d. from $\mathcal{D}$. We are interested in the distance $\|\boldsymbol{x}_k - \boldsymbol{x}_\star\|_2^2$ of the iterates from the unique minimum, and denote the initial distance by $\varepsilon_0 = \|\boldsymbol{x}_0 - \boldsymbol{x}_\star\|_2^2$.

Bach and Moulines [1, Theorem 1] considered this setting[1] and established that

$$k = 2\log(\varepsilon_0/\varepsilon)\Big(\frac{\mathbb{E}L_i^2}{\mu^2} + \frac{\sigma^2}{\mu^2\varepsilon}\Big) \tag{2.2}$$

SGD iterations of the form (2.1), with an appropriate step-size, are sufficient to ensure $\mathbb{E}\|\boldsymbol{x}_k - \boldsymbol{x}_\star\|_2^2 \leq \varepsilon$, where the expectation is over the random sampling. As long as $\sigma^2 = 0$, i.e. the

same minimizer $\boldsymbol{x}_\star$ minimizes all components $f_i(\boldsymbol{x})$ (though of course it need not be a unique minimizer of any of them); this yields linear convergence to $\boldsymbol{x}_\star$, with a graceful degradation as $\sigma^2 > 0$. However, in the linear convergence regime, the number of required iterations scales with the expected *squared* conditioning $\mathbb{E}L_i^2/\mu^2$. In this paper, we reduce this quadratic dependence to a linear dependence. We begin with a guarantee ensuring linear dependence on $\sup L/\mu$:

**Theorem 2.1** *Let each $f_i$ be convex where $\nabla f_i$ has Lipschitz constant $L_i$, with $L_i \leq \sup L$ a.s., and let $F(\boldsymbol{x}) = \mathbb{E}f_i(\boldsymbol{x})$ be $\mu$-strongly convex. Set $\sigma^2 = \mathbb{E}\|\nabla f_i(\boldsymbol{x}_\star)\|_2^2$, where $\boldsymbol{x}_\star = \operatorname{argmin}_{\boldsymbol{x}} F(\boldsymbol{x})$. Suppose that $\gamma \leq 1/\mu$. Then the SGD iterates given by (2.1) satisfy:*

$$\mathbb{E}\|\boldsymbol{x}_k - \boldsymbol{x}_\star\|_2^2 \leq \Big[1 - 2\gamma\mu(1 - \gamma \sup L)\Big]^k \|\boldsymbol{x}_0 - \boldsymbol{x}_\star\|_2^2 + \frac{\gamma\sigma^2}{\mu(1 - \gamma \sup L)}. \qquad (2.3)$$

*That is, for any desired $\varepsilon$, using a step-size of*

$$\gamma = \frac{\mu\varepsilon}{2\varepsilon\mu\sup L + 2\sigma^2} \quad \text{ensures that after} \quad k = 2\log(\varepsilon_0/\varepsilon)\Big(\frac{\sup L}{\mu} + \frac{\sigma^2}{\mu^2\varepsilon}\Big) \qquad (2.4)$$

*SGD iterations, $\mathbb{E}\|\boldsymbol{x}_k - \boldsymbol{x}_\star\|_2^2 \leq \varepsilon$, where $\varepsilon_0 = \|\boldsymbol{x}_0 - \boldsymbol{x}_\star\|_2^2$ and where both expectations are with respect to the sampling of $\{i_k\}$.*

**Proof sketch:** The crux of the improvement over [1] is a tighter recursive equation. Instead of:

$$\|\boldsymbol{x}_{k+1} - \boldsymbol{x}_\star\|_2^2 \leq \big(1 - 2\gamma\mu + 2\gamma^2 L_{i_k}^2\big)\|\boldsymbol{x}_k - \boldsymbol{x}_\star\|_2^2 + 2\gamma^2\sigma^2,$$

we use the co-coercivity Lemma (Lemma A.1 in the supplemental material) to obtain:

$$\|\boldsymbol{x}_{k+1} - \boldsymbol{x}_\star\|_2^2 \leq \big(1 - 2\gamma\mu + 2\gamma^2\mu L_{i_k}\big)\|\boldsymbol{x}_k - \boldsymbol{x}_\star\|_2^2 + 2\gamma^2\sigma^2.$$

The significant difference is that one of the factors of $L_{i_k}$, an upper bound on the second derivative (where $i_k$ is the random index selected in the $k$th iteration) in the third term inside the parenthesis, is replaced by $\mu$, a lower bound on the second derivative of $F$. A complete proof can be found in the supplemental material.

**Comparison to [1]** Our bound (2.4) improves a quadratic dependence on $\mu^2$ to a linear dependence and replaces the dependence on the average squared smoothness $\mathbb{E}L_i^2$ with a linear dependence on the smoothness bound $\sup L$. When all Lipschitz constants $L_i$ are of similar magnitude, this is a quadratic improvement in the number of required iterations. However, when different components $f_i$ have widely different scaling, i.e. $L_i$ are highly variable, the supremum might be significantly larger then the average square conditioning.

**Tightness** Considering the above, one might hope to obtain a linear dependence on the average smoothness $\overline{L}$. However, as the following example shows, this is not possible. Consider a uniform source distribution over $N + 1$ quadratics, with the first quadratic $f_1$ being $N(\boldsymbol{x}[1] - b)^2$ and all others being $\boldsymbol{x}[2]^2$, and $b = \pm 1$. Any method must examine $f_1$ in order to recover $\boldsymbol{x}$ to within error less then one, but by uniformly sampling indices $i$, this takes $N$ iterations in expectation. We can calculate $\sup L = L_1 = 2N$, $\overline{L} = \frac{2(2N-1)}{N}$, $\mathbb{E}L_i^2 = \frac{4(N^2+N-1)}{N}$, and $\mu = 1$. Both $\sup L/\mu = \mathbb{E}L_i^2/\mu^2 = \mathrm{O}(N)$ scale correctly with the expected number of iterations, while error reduction in $\mathrm{O}(\overline{L}/\mu) = \mathrm{O}(1)$ iterations is not possible for this example.

We therefore see that the choice between $\mathbb{E}L_i^2$ and $\sup L$ is unavoidable. In the next Section, we will show how we *can* obtain a linear dependence on the average smoothness $\overline{L}$, using *importance sampling*, i.e. by sampling from a modified distribution.

## 3 Importance Sampling

For a weight function $w(i)$ which assigns a non-negative weight $w(i) \geq 0$ to each index $i$, the weighted distribution $\mathcal{D}^{(w)}$ is defined as the distribution such that

$$\mathbb{P}_{\mathcal{D}^{(w)}}(I) \propto \mathbb{E}_{i \sim \mathcal{D}}\left[1_I(i)w(i)\right],$$

where $I$ is an event (subset of indices) and $1_I(\cdot)$ its indicator function. For a discrete distribution $\mathcal{D}$ with probability mass function $p(i)$ this corresponds to weighting the probabilities to obtain a new probability mass function, which we write as $p^{(w)}(i) \propto w(i)p(i)$. Similarly, for a continuous distribution, this corresponds to multiplying the density by $w(i)$ and renormalizing. Importance sampling has appeared in both the Kaczmarz method [19] and in coordinate-descent methods [14, 15], where the weights are proportional to some power of the Lipschitz constants (of the gradient coordinates). Here we analyze this type of sampling in the context of SGD.

One way to construct $\mathcal{D}^{(w)}$ is through *rejection sampling*: sample $i \sim \mathcal{D}$, and accept with probability $w(i)/W$, for some $W \geq \sup_i w(i)$. Otherwise, reject and continue to re-sample until a suggestion $i$ is accepted. The accepted samples are then distributed according to $\mathcal{D}^{(w)}$.

We use $\mathbb{E}^{(w)}[\cdot] = E_{i \sim \mathcal{D}^{(w)}}[\cdot]$ to denote expectation where indices are sampled from the weighted distribution $\mathcal{D}^{(w)}$. An important property of such an expectation is that for any quantity $X(i)$:

$$\mathbb{E}^{(w)} \left[ \tfrac{1}{w(i)} X(i) \right] = \mathbb{E}\left[ w(i) \right] \cdot \mathbb{E}\left[ X(i) \right], \tag{3.1}$$

where recall that the expectations on the r.h.s. are with respect to $i \sim \mathcal{D}$. In particular, when $\mathbb{E}[w(i)] = 1$, we have that $\mathbb{E}^{(w)} \left[ \tfrac{1}{w(i)} X(i) \right] = \mathbb{E}X(i)$. In fact, we will consider only weights s.t. $\mathbb{E}[w(i)] = 1$, and refer to such weights as *normalized*.

**Reweighted SGD**  For any normalized weight function $w(i)$, we can write:

$$f_i^{(w)}(\boldsymbol{x}) = \frac{1}{w(i)} f_i(\boldsymbol{x}) \quad \text{and} \quad F(\boldsymbol{x}) = \mathbb{E}^{(w)}[f_i^{(w)}(\boldsymbol{x})]. \tag{3.2}$$

This is an equivalent, and equally valid, stochastic representation of the objective $F(\boldsymbol{x})$, and we can just as well base SGD on this representation. In this case, at each iteration we sample $i \sim \mathcal{D}^{(w)}$ and then use $\nabla f_i^{(w)}(\boldsymbol{x}) = \frac{1}{w(i)} \nabla f_i(\boldsymbol{x})$ as an unbiased gradient estimate. SGD iterates based on the representation (3.2), which we will refer to as $w$-weighted SGD, are then given by

$$\boldsymbol{x}_{k+1} \leftarrow \boldsymbol{x}_k - \frac{\gamma}{w(i_k)} \nabla f_{i_k}(\boldsymbol{x}_k) \tag{3.3}$$

where $\{i_k\}$ are drawn i.i.d. from $\mathcal{D}^{(w)}$.

The important observation here is that all SGD guarantees are equally valid for the $w$-weighted updates (3.3)–the objective is the same objective $F(\boldsymbol{x})$, the sub-optimality is the same, and the minimizer $\boldsymbol{x}_\star$ is the same. We do need, however, to calculate the relevant quantities controlling SGD convergence with respect to the modified components $f_i^{(w)}$ and the weighted distribution $\mathcal{D}^{(w)}$.

**Strongly Convex Smooth Optimization using Weighted SGD**  We now return to the analysis of strongly convex smooth optimization and investigate how re-weighting can yield a better guarantee. The Lipschitz constant $L_i^{(w)}$ of each component $f_i^{(w)}$ is now scaled, and we have $L_i^{(w)} = \frac{1}{w(i)} L_i$. The supremum is then given by:

$$\sup L_{(w)} = \sup_i L_i^{(w)} = \sup_i \frac{L_i}{w(i)}. \tag{3.4}$$

It is easy to verify that (3.4) is minimized by the weights

$$w(i) = \frac{L_i}{\overline{L}}, \quad \text{so that} \quad \sup L_{(w)} = \sup_i \frac{L_i}{L_i/\overline{L}} = \overline{L}. \tag{3.5}$$

Before applying Theorem 2.1, we must also calculate:

$$\sigma^2_{(w)} = \mathbb{E}^{(w)}[\|\nabla f_i^{(w)}(\boldsymbol{x}_\star)\|_2^2] = \mathbb{E}[\frac{1}{w(i)}\|\nabla f_i(\boldsymbol{x}_\star)\|_2^2] = \mathbb{E}[\frac{\overline{L}}{L_i}\|\nabla f_i(\boldsymbol{x}_\star)\|_2^2] \leq \frac{\overline{L}}{\inf L}\sigma^2. \tag{3.6}$$

Now, applying Theorem 2.1 to the $w$-weighted SGD iterates (3.3) with weights (3.5), we have that, with an appropriate stepsize,

$$k = 2\log(\varepsilon_0/\varepsilon)\Big(\frac{\sup L_{(w)}}{\mu} + \frac{\sigma^2_{(w)}}{\mu^2\varepsilon}\Big) = 2\log(\varepsilon_0/\varepsilon)\Big(\frac{\overline{L}}{\mu} + \frac{\overline{L}}{\inf L}\cdot\frac{\sigma^2}{\mu^2\varepsilon}\Big) \qquad (3.7)$$

iterations are sufficient for $\mathbb{E}^{(w)}\|\boldsymbol{x}_k - \boldsymbol{x}_\star\|_2^2 \leq \varepsilon$, where $\boldsymbol{x}_\star$, $\mu$ and $\varepsilon_0$ are exactly as in Theorem 2.1.

If $\sigma^2 = 0$, i.e. we are in the "realizable" situation, with true linear convergence, then we also have $\sigma^2_{(w)} = 0$. In this case, we already obtain the desired guarantee: linear convergence with a linear dependence on the average conditioning $\overline{L}/\mu$, strictly improving over the best known results [1]. However, when $\sigma^2 > 0$ we get a dissatisfying scaling of the second term, by a factor of $\overline{L}/\inf L$.

Fortunately, we can easily overcome this factor. To do so, consider sampling from a distribution which is a mixture of the original source distribution and its re-weighting:

$$w(i) = \frac{1}{2} + \frac{1}{2}\cdot\frac{L_i}{\overline{L}}. \qquad (3.8)$$

We refer to this as *partially biased sampling*. Instead of an even mixture as in (3.9), we could also use a mixture with any other constant proportion, i.e. $w(i) = \lambda + (1-\lambda)L_i/\overline{L}$ for $0 < \lambda < 1$. Using these weights, we have

$$\sup L_{(w)} = \sup_i \frac{1}{\frac{1}{2}+\frac{1}{2}\cdot\frac{L_i}{\overline{L}}}L_i \leq 2\overline{L} \quad \text{and} \quad \sigma^2_{(w)} = \mathbb{E}\Big[\frac{1}{\frac{1}{2}+\frac{1}{2}\cdot\frac{L_i}{\overline{L}}}\|\nabla f_i(\boldsymbol{x}_\star)\|_2^2\Big] \leq 2\sigma^2. \quad (3.9)$$

**Corollary 3.1** *Let each $f_i$ be convex where $\nabla f_i$ has Lipschitz constant $L_i$ and let $F(\boldsymbol{x}) = \mathbb{E}_{i\sim\mathcal{D}}[f_i(\boldsymbol{x})]$, where $F(\boldsymbol{x})$ is $\mu$-strongly convex. Set $\sigma^2 = \mathbb{E}\|\nabla f_i(\boldsymbol{x}_\star)\|_2^2$, where $\boldsymbol{x}_\star = \arg\min_{\boldsymbol{x}} F(\boldsymbol{x})$. For any desired $\varepsilon$, using a stepsize of*

$$\gamma = \frac{\mu\varepsilon}{4(\varepsilon\mu\overline{L}+\sigma^2)} \quad \text{ensures that after} \quad k = 4\log(\varepsilon_0/\varepsilon)\Big(\frac{\overline{L}}{\mu} + \frac{\sigma^2}{\mu^2\varepsilon}\Big) \qquad (3.10)$$

*iterations of $w$-weighted SGD (3.3) with weights specified by (3.8), $\mathbb{E}^{(w)}\|\boldsymbol{x}_k - \boldsymbol{x}_\star\|_2^2 \leq \varepsilon$, where $\varepsilon_0 = \|\boldsymbol{x}_0 - \boldsymbol{x}_\star\|_2^2$ and $\overline{L} = \mathbb{E}L_i$.*

This result follows by substituting (3.9) into Theorem 2.1. We now obtain the desired linear scaling on $\overline{L}/\mu$, without introducing any additional factor to the residual term, except for a constant factor. We thus obtain a result which dominates Bach and Moulines (up to a factor of 2) and substantially improves upon it (with a linear rather than quadratic dependence on the conditioning). Such "partially biased weights" are not only an analysis trick, but might indeed improve actual performance over either no weighting or the "fully biased" weights (3.5), as demonstrated in Figure 1.

**Implementing Importance Sampling** In settings where linear systems need to be solved repeatedly, or when the Lipschitz constants are easily computed from the data, it is straightforward to sample by the weighted distribution. However, when we only have sampling access to the source distribution $\mathcal{D}$ (or the implied distribution over gradient estimates), importance sampling might be difficult. In light of the above results, one could use rejection sampling to simulate sampling from $\mathcal{D}^{(w)}$. For the weights (3.5), this can be done by accepting samples with probability proportional to $L_i/\sup L$. The overall probability of accepting a sample is then $\overline{L}/\sup L$, introducing an additional factor of $\sup L/\overline{L}$. This yields a sample complexity with a linear dependence on $\sup L$, as in Theorem 2.1, but a reduction in the number of actual gradient calculations and updates. In even less favorable situations, if Lipschitz constants cannot be bounded for individual components, even importance sampling might not be possible.

## 4  Importance Sampling for SGD in Other Scenarios

In the previous Section, we considered SGD for smooth and strongly convex objectives, and were particularly interested in the regime where the residual $\sigma^2$ is low, and the linear convergence term is dominant. Weighted SGD is useful also in other scenarios, and we now briefly survey them, as well as relate them to our main scenario of interest.

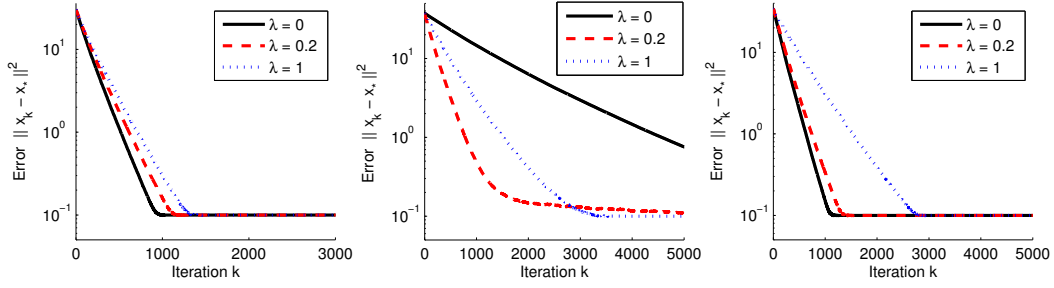

Figure 1: Performance of SGD with weights $w(i) = \lambda + (1 - \lambda)\frac{L_i}{\overline{L}}$ on synthetic *overdetermined* least squares problems of the form (5.1) ($\lambda = 1$ is unweighted, $\lambda = 0$ is fully weighted). Left: $\boldsymbol{a}_i$ are standard spherical Gaussian, $b_i = \langle \boldsymbol{a}_i, \boldsymbol{x}_0 \rangle + \mathcal{N}(0, 0.1^2)$. Center: $\boldsymbol{a}_i$ is spherical Gaussian with variance $i$, $b_i = \langle \boldsymbol{a}_i, \boldsymbol{x}_0 \rangle + \mathcal{N}(0, 20^2)$. Right: $\boldsymbol{a}_i$ is spherical Gaussian with variance $i$, $b_i = \langle \boldsymbol{a}_i, \boldsymbol{x}_0 \rangle + \mathcal{N}(0, 0.1^2)$. In all cases, matrix $\boldsymbol{A}$ with rows $\boldsymbol{a}_i$ is $1000 \times 100$ and the corresponding least squares problem is strongly convex; the stepsize was chosen as in (3.10).

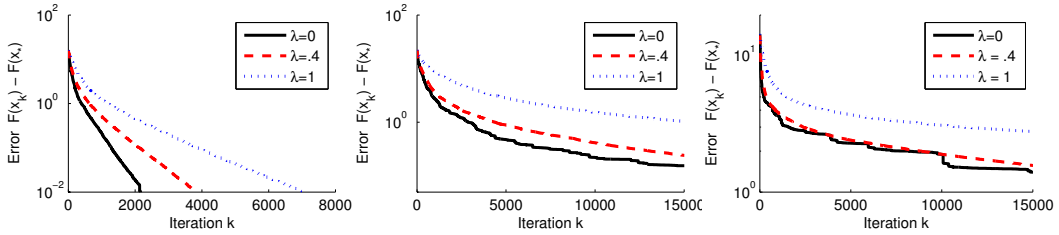

Figure 2: Performance of SGD with weights $w(i) = \lambda + (1 - \lambda)\frac{L_i}{\overline{L}}$ on synthetic *underdetermined* least squares problems of the form (5.1) ($\lambda = 1$ is unweighted, $\lambda = 0$ is fully weighted). We consider 3 cases. Left: $\boldsymbol{a}_i$ are standard spherical Gaussian, $b_i = \langle \boldsymbol{a}_i, \boldsymbol{x}_0 \rangle + \mathcal{N}(0, 0.1^2)$. Center: $\boldsymbol{a}_i$ is spherical Gaussian with variance $i$, $b_i = \langle \boldsymbol{a}_i, \boldsymbol{x}_0 \rangle + \mathcal{N}(0, 20^2)$. Right: $\boldsymbol{a}_i$ is spherical Gaussian with variance $i$, $b_i = \langle \boldsymbol{a}_i, \boldsymbol{x}_0 \rangle + \mathcal{N}(0, 0.1^2)$. In all cases, matrix $\boldsymbol{A}$ with rows $\boldsymbol{a}_i$ is $50 \times 100$ and so the corresponding least squares problem is not strongly convex; the step-size was chosen as in (3.10).

**Smooth, Not Strongly Convex** When each component $f_i$ is convex, non-negative, and has an $L_i$-Lipschitz gradient, but the objective $F(\boldsymbol{x})$ is not necessarily strongly convex, then after

$$k = O\left( \frac{(\sup L)\|\boldsymbol{x}_\star\|_2^2}{\varepsilon} \cdot \frac{F(\boldsymbol{x}_\star) + \varepsilon}{\varepsilon} \right) \tag{4.1}$$

iterations of SGD with an appropriately chosen step-size we will have $F(\overline{\boldsymbol{x}_k}) \leq F(\boldsymbol{x}_\star) + \varepsilon$, where $\overline{\boldsymbol{x}_k}$ is an appropriate averaging of the $k$ iterates [18]. The relevant quantity here determining the iteration complexity is again $\sup L$. Furthermore, the dependence on the supremum is unavoidable and *cannot* be replaced with the average Lipschitz constant $\overline{L}$ [3, 18]: if we sample gradients according to the source distribution $\mathcal{D}$, we must have a linear dependence on $\sup L$.

The only quantity in the bound (4.1) that changes with a re-weighting is $\sup L$—all other quantities ($\|\boldsymbol{x}_\star\|_2^2$, $F(\boldsymbol{x}_\star)$, and the sub-optimality $\varepsilon$) are invariant to re-weightings. We can therefore replace the dependence on $\sup L$ with a dependence on $\sup L_{(w)}$ by using a weighted SGD as in (3.3). As we already calculated, the optimal weights are given by (3.5), and using them we have $\sup L_{(w)} = \overline{L}$. In this case, there is no need for partially biased sampling, and we obtain that

$$k = O\left( \frac{\overline{L}\|\boldsymbol{x}_\star\|_2^2}{\varepsilon} \cdot \frac{F(\boldsymbol{x}_\star) + \varepsilon}{\varepsilon} \right) \tag{4.2}$$

iterations of weighed SGD updates (3.3) using the weights (3.5) suffice. Empirical evidence suggests that this is not a theoretical artifact; full weighted sampling indeed exhibits better convergence rates compared to partially biased sampling in the non-strongly convex setting (see Figure 2), in contrast

to the strongly convex regime (see Figure 1). We again see that using importance sampling allows us to reduce the dependence on $\sup L$, which is unavoidable without biased sampling, to a dependence on $\overline{L}$. An interesting question for further consideration is to what extent importance sampling can also help with stochastic optimization procedures such as SAG [8] and SDCA [17] which achieve faster convergence on finite data sets. Indeed, weighted sampling was shown empirically to achieve faster convergence rates for SAG [16], but theoretical guarantees remain open.

**Non-Smooth Objectives**  We now turn to non-smooth objectives, where the components $f_i$ might not be smooth, but each component is $G_i$-Lipschitz. Roughly speaking, $G_i$ is a bound on the first derivative (the subgradients) of $f_i$, while $L_i$ is a bound on the second derivatives of $f_i$. Here, the performance of SGD (actually stochastic subgradient decent) depends on the second moment $\overline{G^2} = \mathbb{E}[G_i^2]$ [12]. The precise iteration complexity depends on whether the objective is strongly convex or whether $\boldsymbol{x}_\star$ is bounded, but in either case depends linearly on $\overline{G^2}$.

Using weighted SGD, we get linear dependence on

$$\overline{G_{(w)}^2} = \mathbb{E}^{(w)}\left[(G_i^{(w)})^2\right] = \mathbb{E}^{(w)}\left[\frac{G_i^2}{w(i)^2}\right] = \mathbb{E}\left[\frac{G_i^2}{w(i)}\right] \tag{4.3}$$

where $G_i^{(w)} = G_i/w(i)$ is the Lipschitz constant of the scaled $f_i^{(w)}$. This is minimized by the weights $w(i) = G_i/\overline{G}$, where $\overline{G} = \mathbb{E}G_i$, yielding $\overline{G_{(w)}^2} = \overline{G}^2$. Using importance sampling, we therefore reduce the dependence on $\overline{G^2}$ to a dependence on $\overline{G}^2$. Its helpful to recall that $\overline{G^2} = \overline{G}^2 + \mathrm{Var}[G_i]$. What we save is thus exactly the variance of the Lipschitz constants $G_i$.

Parallel work we recently became aware of [22] shows a similar improvement for a non-smooth composite objective. Rather than relying on a specialized analysis as in [22], here we show this follows from SGD analysis applied to different gradient estimates.

**Non-Realizable Regime**  Returning to the smooth and strongly convex setting of Sections 2 and 3, let us consider more carefully the residual term $\sigma^2 = \mathbb{E}\|\nabla f_i(\boldsymbol{x}_\star)\|_2^2$. This quantity depends on the weighting, and in Section 3, we avoided increasing it, introducing partial biasing for this purpose. However, if this is the *dominant* term, we might want to choose weights to minimize this term. The optimal weights here would be proportional to $\|\nabla f_i(\boldsymbol{x}_\star)\|_2$, which is not known in general.

An alternative approach is to bound $\|\nabla f_i(\boldsymbol{x}_\star)\|_2 \leq G_i$ and so $\sigma^2 \leq \overline{G^2}$. Taking this bound, we are back to the same quantity as in the non-smooth case, and the optimal weights are proportional to $G_i$. Note that this differs from using weights proportional to $L_i$, which optimize the linear-convergence term as studied in Section 3.

To understand how weighting according to $G_i$ and $L_i$ are different, consider a generalized linear objective $f_i(\boldsymbol{x}) = \phi_i(\langle \boldsymbol{z}_i, \boldsymbol{x} \rangle)$, where $\phi_i$ is a scalar function with bounded $|\phi_i'|, |\phi_i''|$. We have that $G_i \propto \|\boldsymbol{z}_i\|_2$ while $L_i \propto \|\boldsymbol{z}_i\|_2^2$. Weighting according to (3.5), versus weighting with $w(i) = G_i/\overline{G}$, thus corresponds to weighting according to $\|\boldsymbol{z}_i\|_2^2$ versus $\|\boldsymbol{z}_i\|_2$, and are rather different. E.g., weighting by $L_i \propto \|\boldsymbol{z}_i\|_2^2$ yields $\overline{G_{(w)}^2} = \overline{G^2}$: the same sub-optimal dependence as if no weighting at all were used. A good solution could be to weight by a mixture of $G_i$ and $L_i$, as in the partial weighting scheme of Section 3.

# 5  The least squares case and the Randomized Kaczmarz Method

A special case of interest is the least squares problem, where

$$F(\boldsymbol{x}) = \frac{1}{2}\sum_{i=1}^{n}(\langle \boldsymbol{a}_i, \boldsymbol{x}\rangle - b_i)^2 = \frac{1}{2}\|\boldsymbol{A}\boldsymbol{x} - \boldsymbol{b}\|_2^2 \tag{5.1}$$

with $\boldsymbol{b} \in \mathbb{C}^n$, $\boldsymbol{A}$ an $n \times d$ matrix with rows $\boldsymbol{a}_i$, and $\boldsymbol{x}_\star = \mathrm{argmin}_{\boldsymbol{x}} \frac{1}{2}\|\boldsymbol{A}\boldsymbol{x} - \boldsymbol{b}\|_2^2$ is the least-squares solution. We can also write (5.1) as a stochastic objective, where the source distribution $\mathcal{D}$ is uniform over $\{1, 2, \ldots, n\}$ and $f_i = \frac{n}{2}(\langle \boldsymbol{a}_i, \boldsymbol{x}\rangle - b_i)^2$. In this setting, $\sigma^2 = \|\boldsymbol{A}\boldsymbol{x}_\star - \boldsymbol{b}\|_2^2$ is the residual error

at the least squares solution $x_\star$, which can also be interpreted as noise variance in a linear regression model.

The *randomized Kaczmarz method* introduced for solving the least squares problem (5.1) in the case where $A$ is an overdetermined full-rank matrix, begins with an arbitrary estimate $x_0$, and in the $k$th iteration selects a row $i$ at random from the matrix $A$ and iterates by:

$$x_{k+1} = x_k + c \cdot \frac{b_i - \langle a_i, x_k \rangle}{\|a_i\|_2^2} a_i, \tag{5.2}$$

where $c = 1$ in the standard method. This is almost an SGD update with step-size $\gamma = c/n$, except for the scaling by $\|a_i\|_2^2$.

Strohmer and Vershynin [19] provided the first non-asymptotic convergence rates, showing that drawing rows proportionally to $\|a_i\|_2^2$ leads to provable exponential convergence in expectation [19]. With such a weighting, (5.2) is *exactly* weighted SGD, as in (3.3), with the fully biased weights (3.5).

The reduction of the quadratic dependence on the conditioning to a linear dependence in Theorem 2.1, and the use of biased sampling, was inspired by the analysis of [19]. Indeed, applying Theorem 2.1 to the weighted SGD iterates with weights as in (3.5) and a stepsize of $\gamma = 1$ yields precisely the guarantee of [19]. Furthermore, understanding the randomized Kaczmarz method as SGD, allows us to obtain the following improvements:

**Partially Biased Sampling.** Using partially biased sampling weights (3.8) yields a better dependence on the residual over the fully biased sampling weights (3.5) considered by [19].

**Using Step-sizes.** The randomized Kaczmarz method with weighted sampling exhibits exponential convergence, but only to within a radius, or *convergence horizon*, of the least-squares solution [19, 10]. This is because a step-size of $\gamma = 1$ is used, and so the second term in (2.3) does not vanish. It has been shown [21, 2, 20, 4, 11] that changing the step size can allow for convergence inside of this convergence horizon, but only asymptotically. Our results allow for finite-iteration guarantees with arbitrary step-sizes and can be immediately applied to this setting.

**Uniform Row Selection.** Strohmer and Vershynin's variant of the randomized Kaczmarz method calls for weighted row sampling, and thus requires pre-computing all the row norms. Although certainly possible in some applications, in other cases this might be better avoided. Understanding the randomized Kaczmarz as SGD allows us to apply Theorem 2.1 also with uniform weights (i.e. to the unbiased SGD), and obtain a randomized Kaczmarz using uniform sampling, which converges to the least-squares solution and enjoys finite-iteration guarantees.

## 6  Conclusion

We consider this paper as making three main contributions. First, we improve the dependence on the conditioning for smooth and strongly convex SGD from quadratic to linear. Second, we investigate SGD and importance sampling and show how it can yield improvements not possible without reweighting. Lastly, we make connections between SGD and the randomized Kaczmarz method. This connection along with our new results show that the choice in step-size of the Kaczmarz method offers a tradeoff between convergence rate and horizon and also allows for a convergence bound when the rows are sampled uniformly.

For simplicity, we only considered SGD with fixed step-size $\gamma$, which is appropriate when the target accuracy in known in advance. Our analysis can be adapted also to decaying step-sizes.

Our discussion of importance sampling is limited to a static reweighting of the sampling distribution. A more sophisticated approach would be to update the sampling distribution dynamically as the method progresses, and as we gain more information about the relative importance of components (e.g. about $\|\nabla f_i(x_\star)\|$). Such dynamic sampling is sometimes attempted heuristically, and obtaining a rigorous framework for this would be desirable.

## Footnotes

[1]Bach and Moulines's results are somewhat more general. Their Lipschitz requirement is a bit weaker and more complicated, but in terms of $L_i$ yields (2.2). They also study the use of polynomial decaying step-sizes, but these do not lead to improved runtime if the target accuracy is known ahead of time.

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
