[Supplementary Material]

# A  Proofs

Our main results utilize an elementary fact about smooth functions with Lipschitz continuous gradient, called the co-coercivity of the gradient. We state the lemma and recall its proof for completeness.

## A.1  The Co-coercivity Lemma

**Lemma A.1 (Co-coercivity)** *For a smooth function $f$ whose gradient has Lipschitz constant $L$,*
$$\|\nabla f(\boldsymbol{x}) - \nabla f(\boldsymbol{y})\|_2^2 \le L \left\langle \boldsymbol{x} - \boldsymbol{y}, \nabla f(\boldsymbol{x}) - \nabla f(\boldsymbol{y}) \right\rangle.$$

*Proof.* Since $\nabla f$ has Lipschitz constant $L$, if $\boldsymbol{x}_\star$ is the minimizer of $f$, then
$$\frac{1}{2L}\|\nabla f(\boldsymbol{x}) - \nabla f(\boldsymbol{x}_\star)\|_2^2 = \frac{1}{2L}\|\nabla f(\boldsymbol{x}) - \nabla f(\boldsymbol{x}_\star)\|_2^2 + \langle \boldsymbol{x} - \boldsymbol{x}_\star, \nabla f(\boldsymbol{x}_\star) \rangle \le f(\boldsymbol{x}) - f(\boldsymbol{x}_\star); \tag{A.1}$$
see, for instance, [[13], page 26]. Now define the convex functions
$$G(\boldsymbol{z}) = f(\boldsymbol{z}) - \langle \nabla f(\boldsymbol{x}), \boldsymbol{z} \rangle, \quad \text{and} \quad H(\boldsymbol{z}) = f(\boldsymbol{z}) - \langle \nabla f(\boldsymbol{y}), \boldsymbol{z} \rangle,$$
and observe that both have Lipschitz constants $L$ and minimizers $\boldsymbol{x}$ and $\boldsymbol{y}$, respectively. Applying (A.1) to these functions therefore gives that
$$G(\boldsymbol{x}) \le G(\boldsymbol{y}) - \frac{1}{2L}\|\nabla G(\boldsymbol{y})\|_2^2, \quad \text{and} \quad H(\boldsymbol{y}) \le H(\boldsymbol{x}) - \frac{1}{2L}\|\nabla H(\boldsymbol{y})\|_2^2.$$
By their definitions, this implies that
$$f(\boldsymbol{x}) - \langle \nabla f(\boldsymbol{x}), \boldsymbol{x} \rangle \le f(\boldsymbol{y}) - \langle \nabla f(\boldsymbol{x}), \boldsymbol{y} \rangle - \frac{1}{2L}\|\nabla f(\boldsymbol{y}) - \nabla f(\boldsymbol{x})\|_2^2$$
$$f(\boldsymbol{y}) - \langle \nabla f(\boldsymbol{y}), \boldsymbol{y} \rangle \le f(\boldsymbol{x}) - \langle \nabla f(\boldsymbol{y}), \boldsymbol{x} \rangle - \frac{1}{2L}\|\nabla f(\boldsymbol{x}) - \nabla f(\boldsymbol{y})\|_2^2.$$

Adding these two inequalities and canceling terms yields the desired result.

$\square$

## A.2  Proof of Theorem 2.1

With the notation of Theorem 2.1, and where $i$ is the random index chosen at iteration $k$, we have
$$\begin{aligned}
\|\boldsymbol{x}_{k+1} - \boldsymbol{x}_\star\|_2^2 &= \|\boldsymbol{x}_k - \boldsymbol{x}_\star - \gamma \nabla f_i(\boldsymbol{x}_k)\|_2^2 \\
&= \|(\boldsymbol{x}_k - \boldsymbol{x}_\star) - \gamma(\nabla f_i(\boldsymbol{x}_k) - \nabla f_i(\boldsymbol{x}_\star)) - \gamma \nabla f_i(\boldsymbol{x}_\star)\|_2^2 \\
&= \|\boldsymbol{x}_k - \boldsymbol{x}_\star\|_2^2 - 2\gamma \langle \boldsymbol{x}_k - \boldsymbol{x}_\star, \nabla f_i(\boldsymbol{x}_k) \rangle + \\
&\quad \gamma^2 \|\nabla f_i(\boldsymbol{x}_k) - \nabla f_i(\boldsymbol{x}_\star) + \nabla f_i(\boldsymbol{x}_\star)\|_2^2 \\
&\le \|\boldsymbol{x}_k - \boldsymbol{x}_\star\|_2^2 - 2\gamma \langle \boldsymbol{x}_k - \boldsymbol{x}_\star, \nabla f_i(\boldsymbol{x}_k) \rangle + \\
&\quad 2\gamma^2 \|\nabla f_i(\boldsymbol{x}_k) - \nabla f_i(\boldsymbol{x}_\star)\|_2^2 + 2\gamma^2 \|\nabla f_i(\boldsymbol{x}_\star)\|_2^2 \\
&\le \|\boldsymbol{x}_k - \boldsymbol{x}_\star\|_2^2 - 2\gamma \langle \boldsymbol{x}_k - \boldsymbol{x}_\star, \nabla f_i(\boldsymbol{x}_k) \rangle \\
&\quad + 2\gamma^2 L_i \langle \boldsymbol{x}_k - \boldsymbol{x}_\star, \nabla f_i(\boldsymbol{x}_k) - \nabla f_i(\boldsymbol{x}_\star) \rangle + 2\gamma^2 \|\nabla f_i(\boldsymbol{x}_\star)\|_2^2,
\end{aligned}$$
where we have employed Jensen's inequality in the first inequality and the co-coercivity Lemma A.1 in the final line. We next take an expectation with respect to the choice of $i$. By assumption, $i \sim \mathcal{D}$ such that $F(\boldsymbol{x}) = \mathbb{E}f_i(\boldsymbol{x})$ and $\sigma^2 = \mathbb{E}\|\nabla f_i(\boldsymbol{x}_\star)\|^2$. Then $\mathbb{E}\nabla f_i(\boldsymbol{x}) = \nabla F(\boldsymbol{x})$, and we obtain:
$$\begin{aligned}
\mathbb{E}\|\boldsymbol{x}_{k+1} - \boldsymbol{x}_\star\|_2^2 &\le \|\boldsymbol{x}_k - \boldsymbol{x}_\star\|_2^2 - 2\gamma \langle \boldsymbol{x}_k - \boldsymbol{x}_\star, \nabla \boldsymbol{F}(\boldsymbol{x}_k) \rangle \\
&\quad + 2\gamma^2 \mathbb{E}\left[ L_i \langle \boldsymbol{x}_k - \boldsymbol{x}_\star, \nabla f_i(\boldsymbol{x}_k) - \nabla f_i(\boldsymbol{x}_\star) \rangle \right] + 2\gamma^2 \mathbb{E}\|\nabla f_i(\boldsymbol{x}_\star)\|_2^2 \\
&\le \|\boldsymbol{x}_k - \boldsymbol{x}_\star\|_2^2 - 2\gamma \langle \boldsymbol{x}_k - \boldsymbol{x}_\star, \nabla \boldsymbol{F}(\boldsymbol{x}_k) \rangle \\
&\quad + 2\gamma^2 \sup_i L_i \mathbb{E} \langle \boldsymbol{x}_k - \boldsymbol{x}_\star, \nabla f_i(\boldsymbol{x}_k) - \nabla f_i(\boldsymbol{x}_\star) \rangle + 2\gamma^2 \mathbb{E}\|\nabla f_i(\boldsymbol{x}_\star)\|_2^2 \\
&= \|\boldsymbol{x}_k - \boldsymbol{x}_\star\|_2^2 - 2\gamma \langle \boldsymbol{x}_k - \boldsymbol{x}_\star, \nabla \boldsymbol{F}(\boldsymbol{x}_k) \rangle \\
&\quad + 2\gamma^2 \sup L \langle \boldsymbol{x}_k - \boldsymbol{x}_\star, \nabla \boldsymbol{F}(\boldsymbol{x}_k) - \nabla \boldsymbol{F}(\boldsymbol{x}_\star) \rangle + 2\gamma^2 \sigma^2
\end{aligned}$$

We now utilize the strong convexity of $F(\boldsymbol{x})$ and obtain that

$$\begin{aligned} &\leq \|\boldsymbol{x}_k - \boldsymbol{x}_\star\|_2^2 - 2\gamma\mu(1 - \gamma \sup L)\|\boldsymbol{x}_k - \boldsymbol{x}_\star\|_2^2 + 2\gamma^2\sigma^2 \\ &= (1 - 2\gamma\mu(1 - \gamma \sup L))\|\boldsymbol{x}_k - \boldsymbol{x}_\star\|_2^2 + 2\gamma^2\sigma^2 \end{aligned}$$

when $\gamma\mu \leq 1$. Recursively applying this bound over the first $k$ iterations yields the desired result,

$$\mathbb{E}\|\boldsymbol{x}_k - \boldsymbol{x}_\star\|_2^2 \leq \left(1 - 2\gamma\mu(1 - \gamma \sup L))\right)^k \|\boldsymbol{x}_0 - \boldsymbol{x}_\star\|_2^2 + 2\sum_{j=0}^{k-1}\left(1 - 2\gamma\mu(1 - \gamma \sup L))\right)^j \gamma^2\sigma^2$$

$$\leq \left(1 - 2\gamma\mu(1 - \gamma \sup L))\right)^k \|\boldsymbol{x}_0 - \boldsymbol{x}_\star\|_2^2 + \frac{\gamma\sigma^2}{\mu(1 - \gamma \sup L)}.$$

We next turn to the second part of the theorem, where we optimize the step size $\gamma$ for a fixed tolerance $\varepsilon$. Recall the main recursive step in the previous proof,

$$\mathbb{E}\|\boldsymbol{x}_{k+1} - \boldsymbol{x}_\star\|_2^2 \leq (1 - 2\mu\gamma(1 - \gamma \sup L)) \|\boldsymbol{x}_k - \boldsymbol{x}_\star\|_2^2 + 2\gamma^2\sigma^2, \tag{A.2}$$

which is valid as long as $\mu\gamma \leq 1$. The minimal value of the quadratic

$$F_\xi(\gamma) = (1 - 2\gamma\mu(1 - \gamma \sup L))\,\xi + 2\sigma^2\gamma^2$$

is achieved at

$$\gamma_\xi^* = \frac{\mu\xi}{2\xi\mu \sup L + 2\sigma^2}, \tag{A.3}$$

and

$$F_\xi(\gamma_\xi^*) = \left(1 - \frac{\mu^2\xi}{2\mu \sup L\xi + 2\sigma^2}\right)\xi \tag{A.4}$$

Note that because $\sup L/\mu \geq 1$, it follows that $\mu\gamma_\xi^* \leq 1/2$. Thus if we choose step-size $\gamma^* = \gamma_\varepsilon^*$,

$$\mathbb{E}\|\boldsymbol{x}_{k+1} - \boldsymbol{x}_\star\|_2^2 \leq F_{\|\boldsymbol{x}_k - \boldsymbol{x}_\star\|_2^2}(\gamma^*) \tag{A.5}$$

$$= \left(F_{\|\boldsymbol{x}_k - \boldsymbol{x}_\star\|_2^2}(\gamma^*) - F_\varepsilon(\gamma^*)\right) + F_\varepsilon(\gamma^*) \tag{A.6}$$

$$\leq \left(1 - \frac{\mu^2\varepsilon}{2\mu\varepsilon \sup L + 2\sigma^2}\right)\|\boldsymbol{x}_k - \boldsymbol{x}_\star\|_2^2. \tag{A.7}$$

$$\tag{A.8}$$

Iterating the expectation,

$$\mathbb{E}\|\boldsymbol{x}_{k+1} - \boldsymbol{x}_\star\|_2^2 \leq \left(1 - \frac{\mu^2\varepsilon}{2\mu\varepsilon \sup L + 2\sigma^2}\right)^k \varepsilon_0. \tag{A.9}$$

It follows that if $\varepsilon \leq \mathbb{E}\|\boldsymbol{x}_{k+1} - \boldsymbol{x}_\star\|_2^2$, then

$$\log(\varepsilon/\varepsilon_0) \leq k \log\left(1 - \frac{\mu^2\varepsilon}{2\mu\varepsilon \sup L + 2\sigma^2}\right) \tag{A.10}$$

$$\leq -k\left(\frac{\mu^2\varepsilon}{2\mu \sup L\varepsilon + 2\sigma^2}\right) \tag{A.11}$$

or, equivalently

$$k \leq \log(\varepsilon_0/\varepsilon)\left(\frac{2\mu \sup L\varepsilon + 2\sigma^2}{\mu^2\varepsilon}\right) \tag{A.12}$$

$$= \log(\varepsilon_0/\varepsilon)\left(\frac{2 \sup L}{\mu} + \frac{2\sigma^2}{\mu^2\varepsilon}\right). \tag{A.13}$$

$$\square$$