[Reviews · NeurIPS 2014]

Submitted by Assigned_Reviewer_15

Summary of paper and review:

In this paper, the authors consider stochastic-gradient algorithms, and using
an importance/weighted sampling scheme, they show how it is possible to attain
faster convergence rates in certain regimes. In particular, for strongly
convex problems, the authors show how--if one knows Lipschitz constants of
every term in a finite sum objective--it is possible to attain convergence
rates that depend not on a squared norm of Lipschitz constants but on a
1-norm-like quantity, which is always smaller. The downside of this approach
is that one must know these Lipschitz constants, and it is difficult (perhaps
impossible) to apply the results to objectives that are not of the from f(x) =
\sum_{i=1}^n f_i(x). I am also not convinced that I should care to use these
algorithms; the lack of empirical insights leaves me wondering if this
analysis matters.

Detailed comments:

The idea here is a simple enough idea, and makes sense. In stochastic
gradient algorithms, if one knows the variance/norm of different terms
contributing to the objective, it makes sense to up-sample the most variable
terms and down-sample those that contribute less. This is made clear by
considering the expected norm of the gradient in simple stochastic programming
algorithms; we know that the convergence of these methods is governed by

E[||g||^2] = \int g dP

where g is the stochastic subgradient satisfying E[g] \in \partial f(x). Thus,
if at each iteration we could sample according to a distribution w * dP
instead of simply against dP, we could (by choosing w = ||g|| / \int ||g|| dP)
sample with probabilities w dP and instead use (unbiased) gradients g / w,
which would give a dependence of E[||g||] \le \sqrt{E[||g||^2]}.
The difficulty, of course, is that in most problems it is impossible to
know what the weighting scheme should be, even if upper bounds are known
on the gradients. However, for problems of the form f(x) = \sum_i f_i(x), we
can potentially pre-compute the appropriate constants for each of the terms
f_i in a single pass through the data, thus allowing more efficient sampling.

The biggest challenge of this paper, for me, is the assumption that we always
know all these Lipschitz constants or norms. For linear regression with a
solution, I agree that we can know them via one pass through the data set. But
for many problems--especially in online/stochastic/streaming scenarios--it
will be difficult to either compute these values or use them to do
sampling. The authors made a minor remark that this is "future work," but
it seems a striking weakness.

I would like to see more applications, examples, and experiments than those in
the paper. The analysis, while nice, does not convince me that importance
sampling is truly helpful. Does this matter? Should I try to use this? Do I
have any hope of using it in settings that are large-scale enough that I want
to simply stream the data? Perhaps one natural setting might be in application
to kernel ridge regression; Nystrom sampling (e.g. Bach
http://arxiv.org/abs/1208.2015) and a subsampling approach (COLT 2013,
http://arxiv.org/abs/1305.5029) have claimed benefits, but here we would
simply need to solve (K + \lambda I) \alpha = y, which might be quite fast
with Kaczmarz approaches.

Minor comments:

In general I do not like adverbs, especially editorializing adverbs, in
technical writing. This is a matter of taste, but the first sentence
"remarkably disjoint in the literature" seems to both insult previous
researchers and inflate the chests of the authors. I think the paper speaks
for itself.

There were a few instances with minor imprecise/potentially incorrect
mathematical statements

1. Eq. (2.2), should be \log(\epsilon_0 / \epsilon)

2. Page 2, proof sketch: You use the index i, even though there's randomness
happening; I know it's a sketch, but the imprecision even there is a bit much
(might as well make it correct).

3. Page 3, tightness: The Lipschitz constants I think are L_1 = 1 and L_i =
1/N, as normalized, but perhaps I am confused.

4. Line 540(ish), page 11: I think the stepsizes must satisfy \gamma \le 1/L
and not \gamma \le 1/\mu. (Again, could be misreading.)

5. Eq. (3.7), should be no subscript on the last \sigma^2
Summary: In this paper, the authors consider stochastic-gradient algorithms, and using
an importance/weighted sampling scheme, they show how it is possible to attain
faster convergence rates in certain regimes. In particular, for strongly
convex problems, the authors show how--if one knows Lipschitz constants of
every term in a finite sum objective--it is possible to attain convergence
rates that depend not on a squared norm of Lipschitz constants but on a
1-norm-like quantity, which is always smaller. The downside of this approach
is that one must know these Lipschitz constants, and it is difficult (perhaps
impossible) to apply the results to objectives that are not of the from f(x) =
\sum_{i=1}^n f_i(x). I am also not convinced that I should care to use these
algorithms; the lack of empirical insights leaves me wondering if this
analysis matters.

Submitted by Assigned_Reviewer_42

The paper improves on the dependence on the Lipschitz constants for the stochastic gradient algorithm. It follows the Back/Moulines framework, finds a different recursion, and this leads to a bound using the worst Lipschitz constant rather than the average Lipschitz constant. Then a trivial rescaling sets all Lipschitz constants equal, and the end-result is a better bound. The technical innovation is all in the new recursion bound (which doesn't seem to be tighter, since Back/Moulines was tight, but it is more useful in setting up the rescaling).

The ideas are simple but helpful, and they lead to better bounds, so this is the main strength of the paper.

I would have liked to see more discussion on the practicality of importance sampling and the cost of computing Lipschitz constants. (e.g., in the coordinate descent papers you mention, it is sometimes considered too expensive to find the Lipschitz constants). There was a brief mention of these but without too much investigation. Overall, the paper felt low on material, since the main result was so easy to derive (on the other hand, this does speak well for the clarity of the presentation).

The quadratic improvement over [1] in certain regimes may be true, but I think some of the SG-variants (such as [8]) have the same linear dependence on worst-case condition number.

The claim that this is the first paper to connect Kaczmarz with SGD is a bit ridiculous. You use ideas from [17], but that's it. The connection doesn't seem to go deeper than that, and the fact that minimizing a quadratic is the same as solving a system of equations (or its normal equations) which is clearly not a new observation.

To give the paper more "meat", it would have been interesting to look at batch-sampling and paving results.

Eq (2.2) uses a weird log notation. The first parenthesis is the base of the logarithm? Why not use a sub-script?

Reading page 4, I was concerned about inf_i L_i, but then very happy to see this issue addressed (and resolved) on page 5. Figure 1 is a good illustration (but please add info on the values of n and d used in the simulation).

In section 4, it is reported that there is no need for partially biased sampling. Just as Fig 1 showed that the previous inf L issue was real and not just an artifact, I wonder if the opposite is true in this case. Could you repeat the simulations in Fig 1 for the non-strongly convex case? Just reduce the dimension until it is under-determined.

Non-smooth Objectives. This wasn't clear, since the SGD algorithm is not defined in this case. I assume you replace gradients with subgadients? How to you choose which subgradient to take? If this is standard, please give a reference.

In section 5, please discuss what sigma^2 is here, to give some intuition for the reader. It seems that sigma^2 would be large.

Appendix A.1, I don't see why this is needed since it is so ancillary. You rely on standard analysis results ([12] page 26) in the proof, but lemma A.1 is itself a standard result (Baillon-Haddad theorem), so it doesn't seem to add anything.
Summary: Improves the result of a well-known paper [1] on stochastic gradient descent via a new recursion bound and then applying the appropriate rescaling. Because this seems to be a state-of-the-art result, it's probably worth publishing, but otherwise the paper is a bit light on ideas and doesn't add too much to the literature.

Submitted by Assigned_Reviewer_44

This paper considers stochastic gradient descent, for the problem of minimizing F(x) = \E[f_i(x)]. The paper considers several variants of the problem, depending on whether the functions f_i are smooth, strongly convex, Lipschitz continuous gradient, or just Lipschitz continuous. We discuss these individually more below.

In the introduction, the main point of the paper seems to be the improvement of the rate for linear convergence. Indeed, in the special setting where the optimizer of F is also the optimizer of each individual setting (the authors call this the “realizable setting” which is not very applicable in most problems in machine learning, save for a set of linear equations that are all satisfiable) there is linear convergence. But the paper is nice and has a consistent theme even for less exciting convergence regimes. The authors might consider outlining this more in the introduction, since before I got to the second half of the paper, my impression was that most of the results concerned this very limited regime.

In the first short part of the paper, the authors show that by using a co-coercivity lemma (that gives a bound on the squared difference of the gradient, based on the Lipschitz constant) the dependence in the optimality guarantees in Bach-Moulines on the square of the Lipschitz constants, can be reduced to dependence on the supremum of the Lipschitz constant. While this is a simple result to obtain, using a known result (the co-coercivity lemma) it is a nice observation, and is the starting point for the rest of the paper. In particular, it shows that if the supremum of the Lipschitz constants is not much different than their average, then the dependence on the average Lipschitz constant is reduced from the square appearing in Bach-Moulines, to linear dependence. The supremum, of course, could be significantly bigger than the average. But the main point of this result is that the authors then show that (a) the dependence is real and not a shortcoming of the analysis (it cannot be improved), and that (b) this dependence on sup L_i indicates a natural algorithmic manner of correcting/improving it: importance sampling.

Accordingly, the rest of the paper is about this modification: rather than sample the gradients uniformly, one can sample them according to a different distribution. The authors consider several settings here, including extensions to the non-smooth case, non-strongly convex case, and also the most common

For the smooth and strongly convex case, in the realizable setting, the authors show that the optimal choice is to twist with a distribution proportional to the ratio of Lipschitz constant of the individual function, to the average Lipschitz constant, which is essentially what one would expect from the failure to avoid dependency on the supremum value in the ordinary SGD setting. When, as in any regression setting with noise, the problem is not realizable, this choice comes at a cost. The authors then show that one gets the best of both worlds by doing an averaged sampling, weighting by a mixture of the distribution mentioned above, and the uniform distribution. Using this scheme, the modified SGD algorithm dominates ordinary SGD (and the analysis in Bach-Moulines). Using essentially similar analysis, the authors also consider the non-smooth case, the non-strongly convex case, and the non-realizable case, in each showing that the modified “importance sampling” SGD gives some improvements.

Perhaps the only catch here is that doing this modified sampling may come at a cost (at the very least a cost of sample complexity) depending on how we have access to the samples we draw. There is a short discussion here. Is there something more to say about the additional cost?

Overall, this is a nice paper. The initial focus on the realizable case doesn’t seem to be too compelling. I don’t know of too many cases when we have overdetermined systems where every single equality can be simultaneously satisfied. But there is plenty of interesting material in the rest of the paper.

The paper is generally well written. There are a few very minor typos I found:

46: select —> selects
91: seems like it should be epsilon_0/epsilon instead of the other way around.
119: we the —> we use the
226: get a —> we get a
264: a reduction of in the number of
pp 7 2nd paragraph: then —> than
330: weighing —> weighting
383: rewighting —> reweighting
Summary: The benefits of weighted sampling are interesting -- more interesting probably than the focus on overdetermined yet exactly satisfiable linear equations. The cost of doing weighted sampling might be worthwhile to explore further. Generally a well written paper.
Author Feedback
Author rebuttal: We thank the reviewers for their careful reading of the paper.

The reviewers clearly understood the main contributions (improving the Bach and Moulines bound, and studying the improvement possible through weighted sampling). Below we answer specific questions and address some technical issues raised in the reviews.

Regarding the connection with Kaczmarz: we did not mean to sound arrogant, but only intended to draw attention to this parallel literature---we should have chosen more modest wording.  The fact that solving linear equations and minimizing a quadratic are the same thing is of course very well recognized.  What surprised us is that the Kaczmarz literature, for the most part, analyzes the method directly, and not as a special case of SGD.  This also means that innovations made in the Kaczmarz literature are not directly applicable, as stated, for non-quadratics (the main difference here is quadratics vs non-quadratics).  It also seems that the Kaczmarz literature is not familiar to many researchers working on SGD, and that some working on Kaczmarz do not necessarily realize much of its analysis can be obtained as a special case of SGD analysis.  And so, it would be ridiculous to claim we are the first to realize this connection, but nevertheless, we do think there is value in demonstrating the benefit of making connections between the two bodies of literature.  

Reviewer 1:

* It’s difficult to apply the results to objectives not of the form \sum_{i=1..n} f_i(x)

First, even among such objectives, note that our results do not depend on n (unlike other results that are specific to objectives of this form), and so are applicable even as n->infty.  Second, we do discuss importance sampling, which is applicable even when the objective is not a finite sum (though we agree this might not be entirely satisfying). But it is true that it is easier to see how to apply weighted sampling for finite sums--fortunately, many important objectives are of this form (possibly with additional non-random terms, which we can easily handle).

Typos (thanks for spotting these!):

- Eq (2.2): should indeed be eps_0/eps

- Proof sketch of Theorem 2.1: We will correct i to i_k

- Tightness example : We will fix the normalization of the constants in the example

- Equation (3.7): indeed, second sigma should not have a subscript

Reviewer 2:

* Non-smooth Objectives: SGD isn’t defined.  Do you use sub-gradients? Which subgradient?  Is this standard? Reference?

Yes, as is standard, we use subgradients.  Any subgradient in the subdifferential can be used.  See, e.g. Nemirovski “Efficient Methods in Convex Programming” or Nesterov “Introductory Lectures on Convex Optimization”

* Some of the SG-variants (such as [8]) have the same linear dependence on worst-case condition number.

The comparison with SAG [8] is tricky, since it also involves a dependence on n, which is not present in our analysis (nor in prior analysis of standard SG).

* In Eq. (2.2), is the first parenthesis the base of the log?

The expression log(a)(b+c) just follows standard operator precedence, i.e. the logarithm of a times (b plus c).

* In Section 5, please discuss what sigma^2 is here.

Here, sigma^2 is the residual at the optimal solution (can be interpreted as noise variance in a linear regression model).  We will add a mention of this. We will also add the values of n and d used in Figure 1.

* Why is Appendix A.1 needed?  What’s the point of proving it relying on a citation for Eq. (A.1)?

Since Lemma A.1 is the core of the improvement over [1], we thought it would be nice to include a proof of it in an appendix, instead of sending the reader elsewhere.  Equation (A.1) is essentially just the definition of convexity and Lipschitz continuity---we provided a textbook reference for these, but it should be familiar to most readers without a need to look it up.

* In section 4, it is reported that there is no need for partially biased sampling. Just as Fig 1 showed that the previous inf L issue was real and not just an artifact, I wonder if the opposite is true in this case. Could you repeat the simulations in Fig 1 for the non-strongly convex case? Just reduce the dimension until it is under-determined.

This is a very nice suggestion.  We went ahead and repeated the simulations in Figure 1 for the underdetermined setting.  We found that fully weighted sampling indeed exhibits the fastest convergence rate in all 5 cases considered.

Reviewer 3 :

* The cost of doing weighted sampling might be worthwhile to explore further.

We discussed the cost of weighted sampling via rejection sampling. Once you have all the norms (ie Lipschitz constants) than you can do rejection sampling in O(log(n)) using tree sampling. An interesting but challenging question is: what is a good sampling strategy given only partial information about the norms? Is it possible to learn such information through sampling and re-weight sampling accordingly?

* Minor typos

We will fix the typos, thank you!